# Active and water-soluble form of lipidated Wnt protein is maintained by a serum glycoprotein afamin/α-albumin

Emiko Mihara[1], Hidenori Hirai[1], Hideki Yamamoto[2], Keiko Tamura-Kawakami[1], Mami Matano[3], Akira Kikuchi[2], Toshiro Sato[3], Junichi Takagi[1]*

[1]Laboratory of Protein Synthesis and Expression, Institute for Protein Research, Osaka University, Suita, Japan; [2]Department of Molecular Biology and Biochemistry, Graduate School of Medicine, Osaka University, Suita, Japan; [3]Department of Gastroenterology, Keio University School of Medicine, Tokyo, Japan

**Abstract** Wnt plays important role during development and in various diseases. Because Wnts are lipidated and highly hydrophobic, they can only be purified in the presence of detergents, limiting their use in various in vitro and in vivo assays. We purified N-terminally tagged recombinant Wnt3a secreted from cells and accidentally discovered that Wnt3a co-purified with a glycoprotein afamin derived from the bovine serum included in the media. Wnt3a forms a 1:1 complex with afamin, which remains soluble in aqueous buffer after isolation, and can induce signaling in various cellular systems including the intestical stem cell growth assay. By co-expressing with afamin, biologically active afamin-Wnt complex can be easily obtained in large quantity. As afamin can also solubilize Wnt5a, Wnt3, and many more Wnt subtypes, afamin complexation will open a way to put various Wnt ligands and their signaling mechanisms under a thorough biochemical scrutiny that had been difficult for years.

*For correspondence: takagi@ protein.osaka-u.ac.jp

**Competing interests:** The authors declare that no competing interests exist.

## Introduction

Wnt proteins constitute a large family of secreted glycoproteins that control diverse aspects of embryonic development and adult homeostasis (*Logan and Nusse, 2004*). As they influence the balance between proliferation and differentiation in many cell types, they are fundamentally implicated in the biological processes with great medical importances, including bone formation (*Regard et al., 2012*), immune regulation (*Yu et al., 2010*), cancer (*Zimmerman et al., 2012*), and stem cell renewal (*ten Berge et al., 2011*). At least 19 Wnt proteins are present in mammals, each serving different but potentially overlapping functions (*Holstein, 2012*). All mammalian Wnts are predicted to be covalently lipidated at a conserved Ser residue (*Takada et al., 2006*; *Willert et al., 2003*), a modification essential to their biological activity because of its contribution to the interaction with Frizzled receptors (*Janda et al., 2012*). Due to the strong hydrophobic property granted by the lipid moiety, however, purified Wnt proteins tend to lose their biological activity, probably due to the aggregation even in the presence of detergents (*Dhamdhere et al., 2014*).

The most general source for soluble Wnt ligands used in a paracrine-type experiments (i.e., exogenous addition of ligands) is the culture supernatants of cells producing Wnts. In fact, L cells (mouse fibroblastic cell line) stably expressing Wnt3a (*Willert et al., 2003*) or Wnt5a (*Chen et al., 2003*) have been established and are already available from a public cell bank. Curiously, it has been known that Wnt secretion from cultured cells require the presence of bovine serum in the media (*Willert, 2008*). As a result, Wnt-containing culture media would always have to contain ~10% serum, precluding the use of such sample in biological assays that are sensitive to the factors contained in

**eLife digest** The Wnt signaling pathway helps animal cells to communicate with each other to coordinate the formation of tissues and organs. The pathway relies on a protein called Wnt that is released from cells and binds to a receptor protein called Frizzled on the surface of other cells to trigger changes in gene activation. Defects in the Wnt signaling pathway contribute to cancer and other diseases.

Great progress has been made in understanding Wnt signaling, but certain types of experiments have been hindered because it has been difficult to isolate pure Wnt proteins. This is partly because Wnt proteins are attached to a fatty molecule that is important for their activity but also makes these proteins "hydrophobic," or repelled by water.

Hydrophobic proteins have a strong tendency to clump or aggregate when they are isolated from cells, which reduces the biological activity of proteins. Adding detergents to the aggregates can break them apart, but can also hinder the proteins' activities and cannot be used in all experiments. Previous research has shown that mammalian cells grown in the presence of blood serum can produce Wnt proteins that do not aggregate. Blood serum is a complex mixture of different molecules obtained from blood and is commonly added to cells grown in the laboratory. However, adding serum can have also undesirable effects and it is not understood why serum stops Wnt proteins forming aggregates.

Using biochemical methods, Mihara et al. have now identified the component in blood serum that prevents Wnt proteins from aggregating. The experiments showed that a protein in the blood serum called afamin binds tightly to Wnt proteins. Furthermore, the complex between afamin and Wnt was biologically active, and could bind to the Frizzled receptor and trigger an appropriate response in cells.

Mihara et al. then generated cells that produced both afamin and Wnt and used them to purify large amounts of biologically active Wnt/afamin complexes. This method avoids the potentially undesirable effects of using detergents or serum, and will therefore likely be useful for future experiments and therapeutic applications. Further work is also needed to understand why afamin binds to Wnt proteins and whether this is important for Wnt signaling.

bovine serum (e.g., growth factors and proteases). The serum component(s) that support Wnt secretion has not been identified, and it is generally assumed that serum lipids render the hydrophobic Wnt proteins soluble. Willert et al had found that the inclusion of a detergent CHAPS at 1% can solubilize Wnt proteins and successfully developed a method to purify Wnt proteins from serum-containing media, which opened a way to apply purified Wnts to various experimental systems (*Willert, 2008*). By using this technology, several Wnt ligands are now commercially available in a purified format. However, the Wnt ligands thus purified contain CHAPS and are incompatible with detergent-sensitive cell-based assays. Moreover, CHAPS-solubilized Wnt quickly loses its biological activity upon incubation at 37°C, unless it is reconstituted into lipid vesicle (*Dhamdhere et al., 2014*).

In the present report, we identified the serum component responsible for the aqueous partitioning of Wnt proteins as a serum glycoprotein afamin rather than lipid molecules. Afamin and Wnt apparently makes 1:1 complex in the culture media, which is highly soluble in detergent-free buffers upon isolation but is decomposed upon the addition of 1% CHAPS. Afamin-Wnt complex is biologically active, indicating that afamin can deliver Wnt ligands to its receptors on cell surface. By using afamin as a carrier, we succeeded in preparing large quantity of biologically active Wnt3a, Wnt5a, and Wnt3 proteins with no other components, which can be stored at 4°C.

## Results

We first systematically explored the tagging condition for the expression and secretion of mouse Wnt3a in HEK cells, and found that N-terminal addition of the 21-residue 'TARGET(Tg)' tag developed in our lab (*Tabata et al., 2010*) (see schematics in *Supplementary file 1*) was compatible. As shown in *Figure 1A*, conditioned medium (CM) from HEK cells stably expressing Tg-Wnt3a

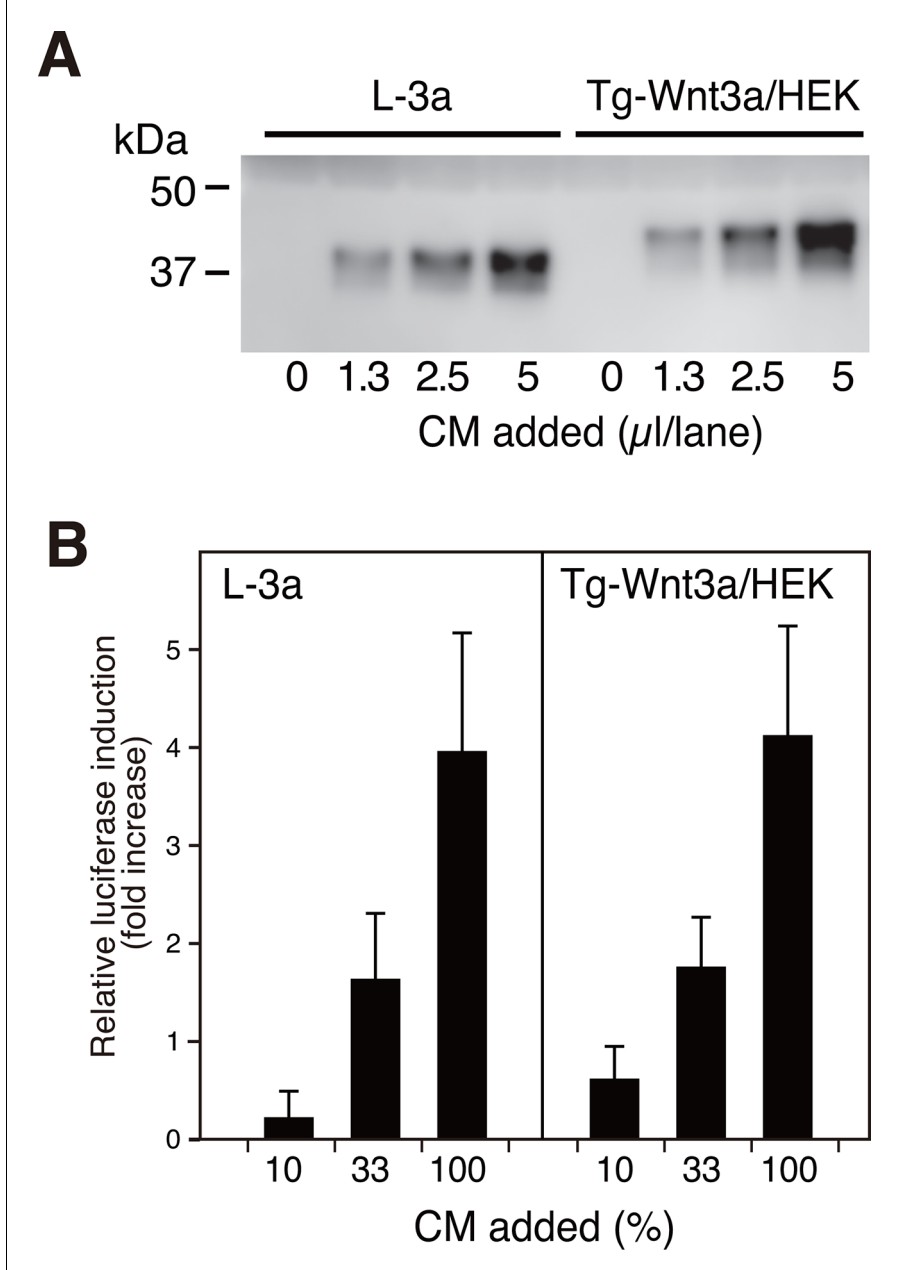

**Figure 1.** N-terminally tagged Wnt3a secretion from HEK cells. (**A**) Indicated amounts of CM from the confluent L cells stably expressing untagged Wnt3a (L-3a) or HEK293S GnT1⁻ cells stably expressing TARGET-tagged Wnt3a (Tg-Wnt3a/HEK) were subjected to a Western blotting using anti-mouse Wnt3a antibody. Note that Tg-Wnt3a migrate slower than the untagged Wnt3a due to the presence of extra 35-residue (~4 kDa) tag sequence. (**B**) The stable TCF reporter cells were incubated with the indicated concentration of CM for 6 hr. Luciferase activities in the cell lysates were determined and expressed as the relative increase from the control value obtained in the mock-treated cells. Data are mean ± SD of three independent experiments, in which quadruplicate determinations were made. See also **Figure 1—source data 1**.

The following source data is available for figure 1:

**Source data 1.** The Excel spreadsheet source file for **Figure 1B**.

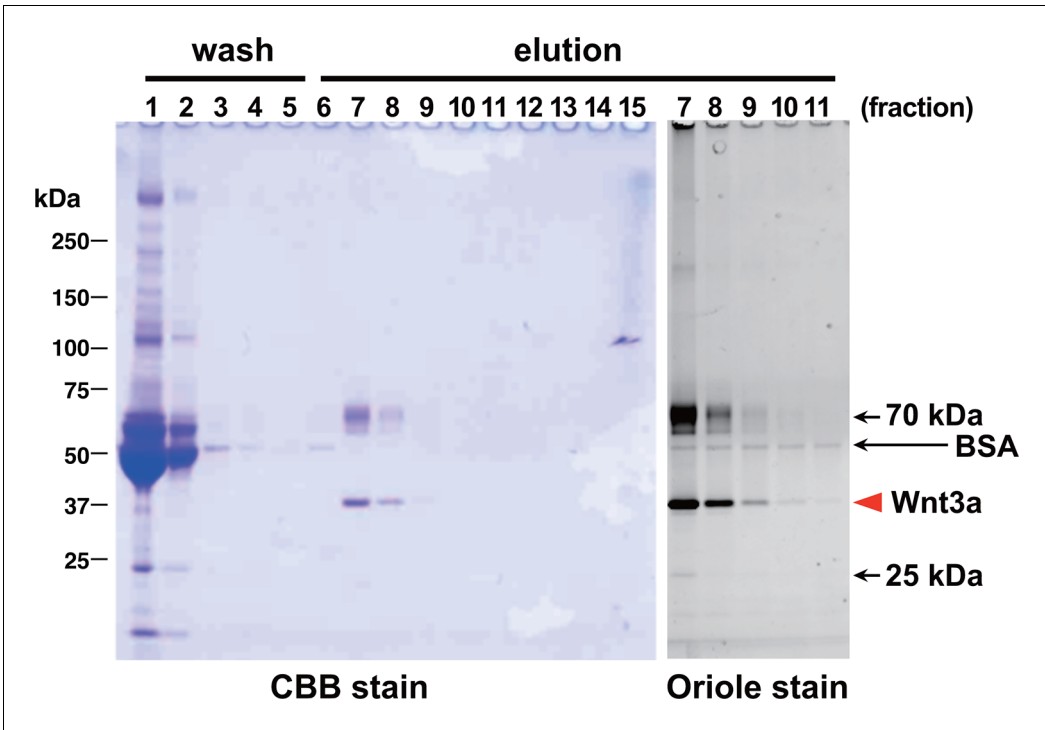

**Figure 2.** Affinity purification of tagged Wnt3a from the culture supernatants. A small column of Sepharose coupled with anti-TARGET tag antibody P20.1 (~3 ml) that had been incubated with ~220 ml of CM from the confluent culture was washed with TBS, followed by elution with TBS containing 0.2 mg/ml competing C8 peptide. Ten µl sample from each fraction (fraction size = 3 ml) was subjected to 5–20% SDS-PAGE under nonreducing condition and stained with Coomassie Blue. A portion of the same gel (fractions 7–11) was stained with Oriole fluorescent stain to visualize minor contaminating bands.

contained Wnt3a antigen at a level comparable to that from L-3a cells. The Wnt reporter assay revealed that both CM possessed similar activity (*Figure 1B*), indicating that the N-terminal tagging did not impair the biological activity of Wnt3a that had been successfully secreted into the media. The TARGET tag system takes advantage of the specific recognition of concatenated YPGQ sequence by a monoclonal antibody P20.1 (*Nogi et al., 2008*), allowing the rapid one-step purification of tagged protein by a buffer containing competing peptide or 40% propylene glycol. Affinity purification of Tg-Wnt3a from the CM was conducted on P20.1-Sepharose, resulting in the ~37 kDa protein band corresponding to the tagged Wnt3a appearing in the peptide-eluted fractions (*Figure 2*, lanes 7–9). Near-complete removal of the bovine serum albumin (BSA), the largest protein constituent present in the CM, was achieved during the purification, although very small amount of BSA is still contaminated in the eluted fractions. However, we noticed that protein bands with relative molecular masses of 70 and 25 kDa were always present in the fractions containing Wnt3a (*Figure 2*). Among them, the 70-kDa protein was reproducibly co-purified with Wnt3a at near stoicheometric fashion, while the amount of 25-kDa protein was generally much lower than Wnt3a and varied among different experiments. As these proteins do not appear in the elution fractions from the P20.1-Sepharose incubated with the control CM lacking Wnt3a, they are likely to associate and be co-purified with Wnt3a. The N-terminal sequence of the 25-kDa band (GDDPQSSWDRV) revealed that it is the bovine apolipoprotein A1 (ApoA1; NP_776667). This is consistent with the report by Neumann et al that a part of secreted Wnt3a is released onto ApoA1-containing HDL particles (*Neumann et al., 2009*). In contrast, N-terminal Edman sequencing of the 70-kDa band derived a sequence of LPTQPQDVDD, which was in 100% match with the first 10 residues of a relatively unknown protein called afamin (NP_001179104).

Afamin (AFM), also known as α-albumin, is a plasma glycoprotein that belongs to the albumin superfamily proteins (*Lichenstein et al., 1994*). Little is known about the physiological function of

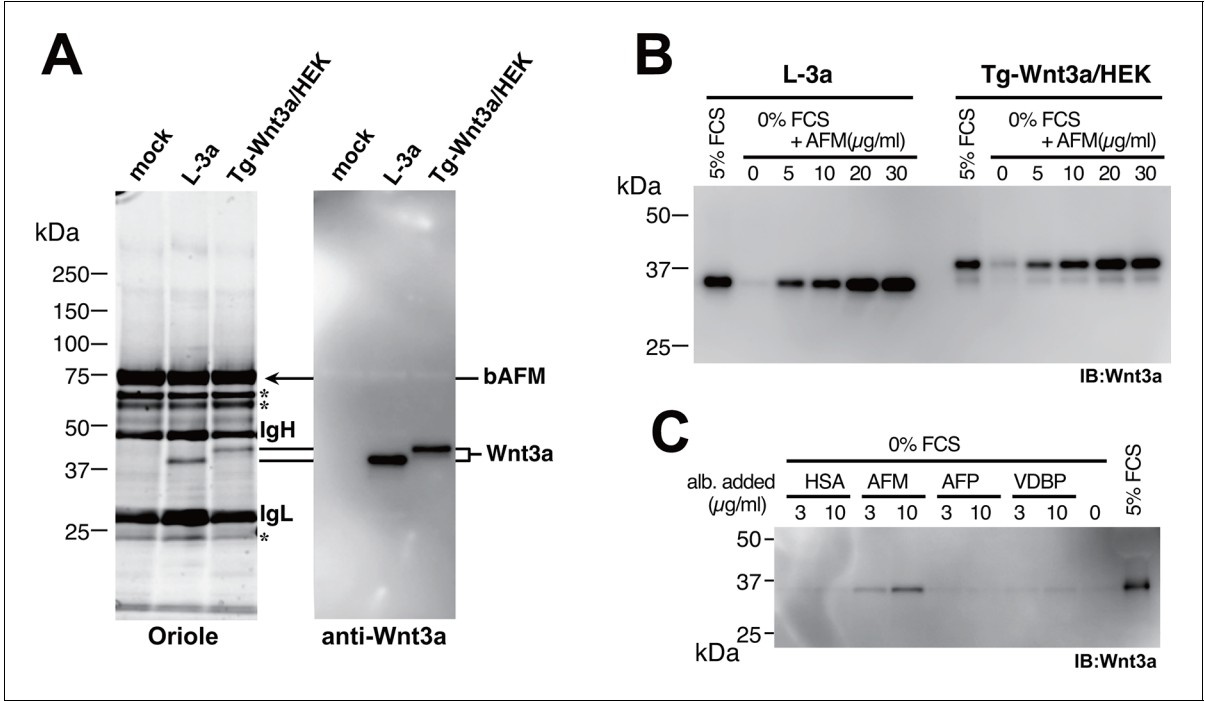

**Figure 3.** Serum afamin binds to Wnt3a secreted from cells. (**A**) Serum-containing CM from the mock, L-3a, or Tg-Wnt3a/HEK cells were incubated with Sepharose beads immobilized with an anti-bovine AFM monoclonal antibody (clone B91) and the immunoprecipitated materials were separated by SDS-PAGE under reducing condition, followed by Oriole Fluorescent protein stain (left) and anti-Wnt3a immunoblotting (right). Positions for bovine AFM and tagged/untagged Wnt3a are shown in the right. Asterisks denote nonspecific serum-derived proteins. (**B**) L-3a or Tg-Wnt3a/HEK cells were cultured in DMEM containing various concentrations of serum or purified recombinant human AFM for 5 days, and the resultant CMs (3 μl) were subjected to the anti-Wnt3a immunoblotting. (**C**) Effect of various albumin family proteins to support Wnt3a secretion from L-3a cells is evaluated as in (**B**). HSA, human serum albumin; AFM, human afamin; AFP, human α-fetoprotein; VDBP, mouse vitamin D binding protein.

The following figure supplement is available for figure 3:

**Figure supplement 1.** Purified recombinant albumin family proteins.

this protein except for its potential role as a carrier of vitamin E (*Kratzer et al., 2009*), and there have been no reports that suggest functional linkage between AFM and Wnts. In order to confirm that AFM and Wnt3a are associated in the CM, we raised a monoclonal antibody against bovine AFM and performed immunoprecipitation from the Wnt3a-containing CM. As clearly shown in *Figure 3A*, both tagged and untagged Wnt3a can be co-immunoprecipaed with bovine AFM from the CM, indicating that Wnt3a itself but not the tag portion has ability to specifically bind to AFM. This result prompted us to investigate whether serum AFM represents the agent responsible for the ability of serum to allow soluble Wnt3a secretion into the medium. To this end, we produced and purified recombinant human AFM protein using mammalian expression system (*Figure 3—figure supplement 1*) and tested its ability to facilitate Wnt3a secretion into the serum-free media by L-3a and stable Tg-Wnt3a-expressing HEK cells. As reported previously, Wnt3a was successfully secreted into the media when 5% fetal calf serum (FCS) is present, but is nearly absent in the supernatant when the same cells are cultured in a plain DMEM (*Figure 3B*). Remarkably, addition of purified recombinant human AFM into the media restored the Wnt3a secretion in a dose dependent manner, even though the medium contained no other proteins or special additives (*Figure 3B*). Albumin superfamily comprises of four members, including serum albumin, AFM, α-fetoprotein (AFP), and vitamin D binding protein (VDBP), all present in blood at different concentrations (*Lichenstein et al., 1994*). We prepared all of them in recombinant form (*Figure 3—figure supplement 1*) and tested their ability to support Wnt3a secretion. As shown in *Figure 3C*, only AFM was able to enhance Wnt3a secretion, suggesting that AFM is the major serum factor responsible for this activity.

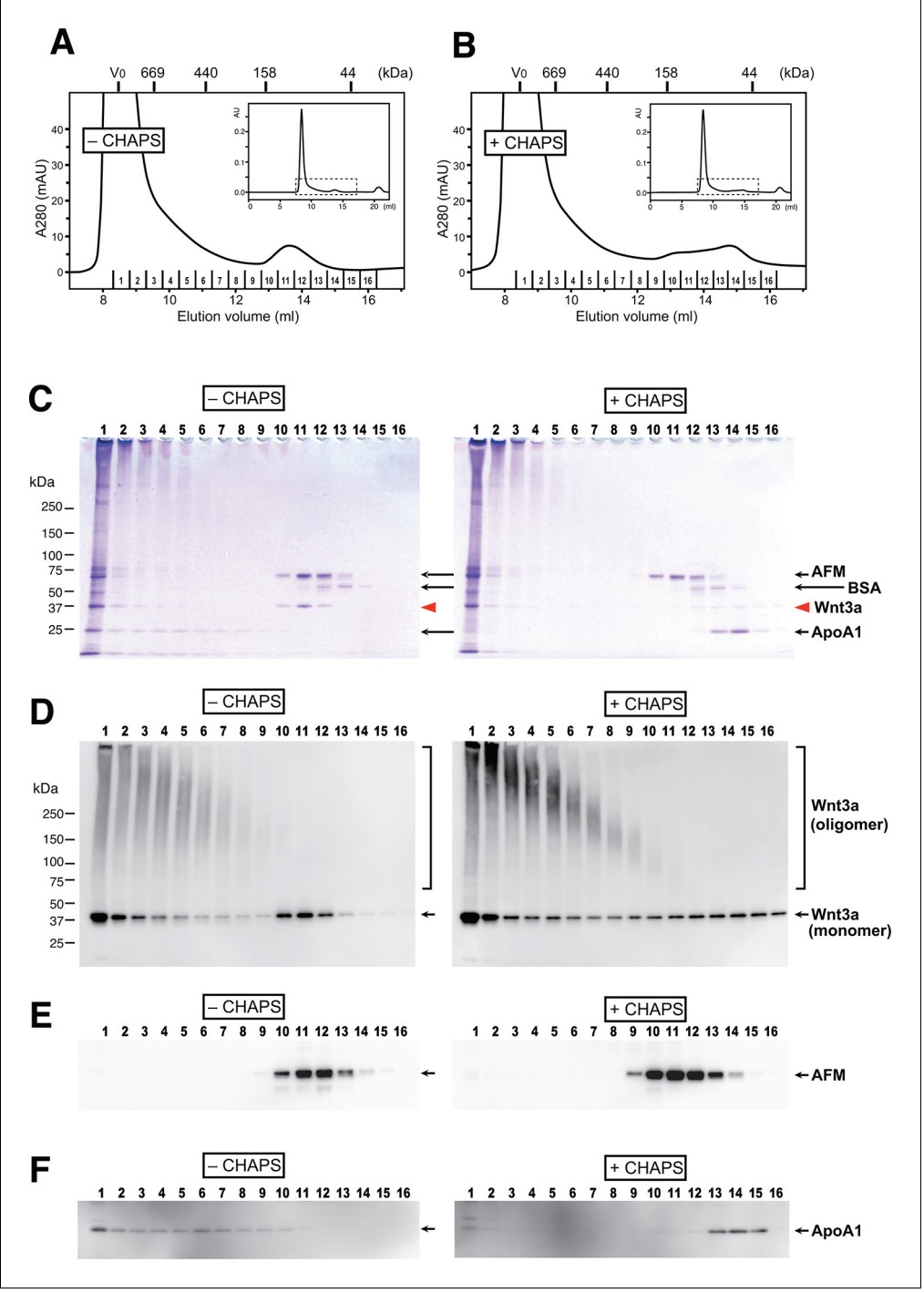

**Figure 4.** AFM and Wnt3a form stable 1:1 complex in the absence of detergents. (**A,B**) SEC profiles of affinity-purified Wnt3a preparation in the absence (**A**, –CHAPS) or presence (**B**, +CHAPS) of 1% CHAPS. Graphs are expanded view of the areas indicated by dotted line box in the whole chromatograms (*insets*). Elusion positions for molecular mass standards including thyroglobulin (669 kDa), ferritin (440 kDa), aldorase (158 kDa) and ovalbumin (44 kDa) are indicated at the *top*. Sixteen fractions collected from each chromatography are subjected to nonreducing SDS-PAGE on 5–20% gradient gels, followed by Coomassie Blue staining (**C**) or immunoblotting with anti-Wnt3a (**D**), anti-bovine AFM (**E**), or anti-bovine ApoA1 (**F**). In (**C**) ~ (**F**), analysis of the samples from (**A**) and (**B**) are shown in the left (–CHAPS) or right (+CHAPS) panels, respectively.

Although many researchers have purified Wnt proteins from serum-containing culture media, no one had reported the presence of serum afamin in their preparations. As the critical trick for the successful Wnt purification discovered by Willert et al is the addition of 1% CHAPS at the beginning of the purification steps (*Willert, 2008*), we speculated that AFM-Wnt interaction may be susceptible to a detergent treatment. In order to test this hypothesis, the partially pure Wnt3a preparation containing AFM was subjected to size exclusion chromatography (SEC) on a Superdex 200 column in the presence or absence of 1% CHAPS. Under both conditions, the majority of the proteins formed large high-molecular weight (HMW) aggregates eluting at a void peak around 8–10 ml (*Figures 4A and B*). In the detergent-free condition (panels labeled as "– CHAPS" in *Figure 4*), however, there is a second peak eluting at 13.6 ml corresponding to the size of ~100 kDa. Coomassie staining (*Figure 4C*) and anti-Wnt3a immunoblot (*Figure 4D*) revealed that Wnt3a protein is contained in this 100-kDa peak as well as in the HMW fractions. When the same set of samples were probed with anti-AFM antisera, AFM immunoreactivity was solely present in the 100-kDa peak and not in the HMW fractions (*Figure 4E*). As the apparent size of this peak (100 kDa) is close to the sum of the molecular masses of Wnt3a (37 kDa) and AFM (70 kDa), we conclude that this peak represents the stable and monodispersed 1:1 complex formed between Wnt3a and AFM. In contrast to AFM, the 25-kDa ApoA1 antigen was broadly present in the HMW fractions but not in the 13.6-ml peak (*Figure 4F*), suggesting that the Wnt3a trapped in HDL particles accounts for at least part of the HMW peak. It also became evident that the HMW fractions are enriched by Wnt3a oligomers that are covalently crosslinked by disulfide bonds, while such oligomers are absent in AFM-containing fractions (*Figure 4D*). In sharp contrast to the results described above, the gel filtration profile obtained in the presence of 1% CHAPS (panels labeled as +CHAPS) shows completely different pattern; the Wnt3a do not co-elute with AFM any more but present in a broad range of fractions centered around ~50 kDa, with a concomitant increase in the covalent oligomer formation (*Figures 4B–D*). In addition, elution positions for ApoA1 is shifted to the fractions with much smaller size (*Figure 4F*), suggesting that the HDL particles can no longer exist under this condition. Based on these observations, we conclude that the specific interaction between Wnt3a and AFM is lost in the presence of 1% CHAPS, hence evaded from the discovery of its presence in the previous studies.

We next checked if the Wnt3a complexed with AFM retains its biological activity. When the purified AFM-Wnt3a complex dissolved in a CHAPS-free buffer was tested in the Wnt/β-catenin-responsive TCF reporter assay using HEK293 cells stably expressing luciferase gene under the control of TCF/LEF, concentration-dependent induction of reporter was observed (*Figure 5A*, left). The positive signal was detected at concentration as low as 0.56 nM (20 ng/ml), and reached the saturation at around 15 nM (500 ng/ml). In contrast, commercially available carrier-free Wnt3a protein (R&D Systems, purified according to the method by Willert et al. (*Willert, 2008*)) exhibited much lower activity in the same assay condition (*Figure 5A*, right). Because of the concentration range employed for this sample (<45 nM) to minimize the amount of CHAPS brought into the cell culture condition, it is not clear if the signal reached the saturation. Assuming that the maximum reporter induction achievable by two different Wnt3a preparations is identical, the EC50 value for the AFM-Wnt3a complex is estimated to be 5~10-fold lower than that for the purified Wnt3a. Furthermore, AFM-Wnt3a complex exhibited similar activity before and after the storage at 4°C for 1 month in PBS (*Figure 5—figure supplement 1*), indicating the stable nature of this preparation in the absence of any detergents.

The biological activity of AFM-Wnt3a complex was also evaluated using the human intestinal organoid culture. Active Wnt3a signal is required for the stem cells present at the bottom of the intestinal crypt to self-renew (*Sato and Clevers, 2013*), and it is established that human colonic organoids require a set of growth factors (including Wnt3a, EGF, noggin, and R-spondin-1) for sustained expansion (*Sato et al., 2011*). Incubation of the single-cell dissociated human intestinal organoids with the serum-containing CM of Wnt3a-expressing cells together with other components successfully supported the continued growth and expansion of the organoids (*Figure 5B*, 'Wnt3a CM), while in the absence of Wnt3a (*Figure 5B*, control) the cells ceased proliferation after the second passage. However, purified Wnt3a obtained from a commercial source failed to show this activity when added at 300 ng/ml to the culture (*Figure 5B*, Wnt3a(R&D)). This result is consistent with the finding by Dhamdhere et al that CHAPS-purified Wnt3a lose its activity upon prolonged incubation at 37°C (*Dhamdhere et al., 2014*). It seems likely that continued presence of active Wnt3a during the long culture (7 days per passage) is required for the expansion of the organoid. In contrast,

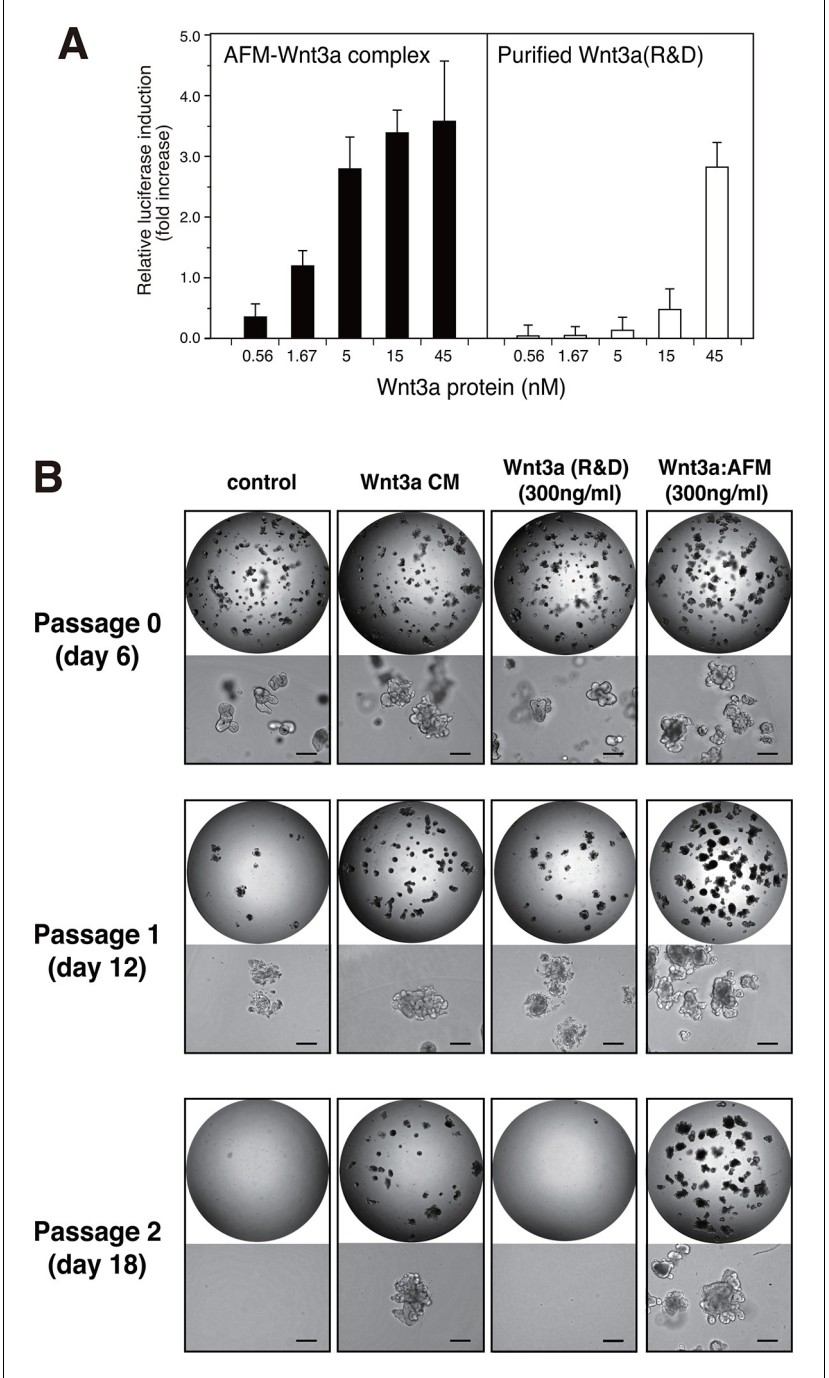

**Figure 5.** Wnt3a in complex with AFM is biologically active. (A) Comparison between the purified Wnt3a from a commercial source and the AFM-Wnt3a complex. Increasing concentrations of Wnt3a proteins are added to TCF reporter cells and the Wnt signaling activities are evaluated as in *Figure 1B*. The data are mean ± SD of three independent experiments, in which quadruplicate determinations were made. See *Figure 5—source data 1*. (B) AFM-Wnt3a complex supports self-renewal of human intestinal stem cells. The ability of single-cell dissociated human intestinal organoids to expand was evaluated by culturing in the absence (control) or presence of various forms of Wnt3a preparations. Photographs are taken every 7 days at a low (x2, entire view of one 48-well, upper panels) and the high (x 10, lower panels) magnifications. Bar: 200 µm.

The following source data and figure supplements are available for figure 5:

**Source data 1.** The Excel spreadsheet source file for *Figure 5A*.

*Figure 5 continued on next page*

*Figure 5 continued*

**Source data 2.** The Excel spreadsheet source file for *Figure 5—figure supplement 1*.

**Source data 3.** The Excel spreadsheet source file for *Figure 5—figure supplement 2*.

**Figure supplement 1.** Stability of the AFM-Wnt3a complex.

**Figure supplement 2.** Quantitative analysis of the human intestinal organoid growth and expansion.

AFM-Wnt3a complex was able to support organoid expansion at the same concentration (*Figure 5B*, Wnt3a:AFM, and *Figure 5—figure supplement 2*). When supplemented with the same concentration of AFM-Wnt3a complex in the medium for each passage, organoids were able to expand for more than 18 passages over a period of 4 months (data not shown). As this culture media does not include any detergents, high salt, or serum components, it provides ideal experimental condition to evaluate growth, renewal, and differentiation of the intestinal or other type of stem cells for various studies.

To check if the ability of AFM to solubilize and stabilize Wnt protein in aqueous environment is specific to Wnt3a, we established a stable HEK cell line expressing N-terminally tagged mouse Wnt5a and purified it from the CM. As shown in *Figure 6A*, Tg-Wnt5a co-purified with serum afamin by the anti-tag immunoaffinity chromatography, just like Wnt3a (compare with *Figure 2*). Furthermore, the SEC profile of the immunopurified Wnt5a preparation was essentially the same as that of Wnt3a, showing the presence of monodispersed AFM-Wnt5a complex peak eluting at around 150 kDa, in addition to the HMW aggregate fractions (*Figure 6B* and *Figure 6—figure supplement 1*). Finally, the AFM-Wnt5a complex thus purified was biologically active, because it induced phosphorylation of Dvl2 in NIH3T3 cells at concentrations above 10 ng/ml (*Figure 6C*). Importantly, the activity of AFM-Wnt5a complex was indistinguishable with that of AFM-free Wnt5a protein freshly purified from the CM using the conventional method (*Kurayoshi et al., 2007*).

As an exogenous addition of AFM in the serum-free culture condition promoted secretion of Wnt3a from cells (*Figure 3B*), we hypothesized that AFM may be able to enhance the production of Wnts when present during their biosynthesis within cells. Therefore, we evaluated the effect of AFM co-expression on the transient expression of Wnt3a in Expi293F cells. In order to increase the yield during the immunoaffinity purification, we changed the tag sequence attached to the N-terminus of Wnt3a to a recently developed PA tag (construct #3 in *Supplementary file 1*), which has shorter sequence than the TARGET tag and yet recognized by its antibody (NZ-1) with very high affinity (*Fujii et al., 2014*). Transient transfection of PA-Wnt3a alone into the suspension culture of Expi293F cells did not result in the secretion of Wnt3a (*Figure 7A*. lane 1), which was expected because the culture medium for this cell line did not contain serum. In contrast, when the cells were co-transfected with PA-Wnt3a and human AFM, ~37-kDa Wnt3a band became visible in the NZ-1 immuno-precipitates along with the associated AFM (*Figure 7A*, lane 2), confirming the ability of AFM to enhance Wnt production/secretion by co-expression. Using this co-expression system, we further evaluated the effect of AFM on the expression and secretion of various Wnt family proteins other than mouse Wnt3a and Wnt5a. To this end, expression constructs for all 19 members of human Wnt proteins containing N-terminal PA tag were prepared from the Open Source Wnt Project DNA collection (*Najdi et al., 2012*). Out of the 19 PA-tagged human Wnts, 12 members (Wnt1, Wnt2b, Wnt3, Wnt3a, Wnt5a, Wnt7a, Wnt7b, Wnt8a, Wnt9a, Wnt9b, Wnt10a, and Wnt10b) could be secreted at detectable level from HEK293T cells when transfected alone in the presence of serum (*Figure 7—figure supplement 1*). These 12 Wnts were then subjected to the AFM co-expression experiments. As in the case of mouse Wnt3a, 12 human Wnt proteins exhibited no or very little secretion into the serum-free culture media when singly transfected into the Expi293F cells (*Figure 7B*, lanes marked with –). However, their expression was enhanced upon the co-transfection with human AFM (*Figure 7B*, lanes marked with +). Although the extent of the enhancement varied among different Wnts, all Wnts directly associated with AFM as evidenced by the co-immunoprecipitation experiments (*Figure 7C*), suggesting that AFM can serve as a secretion vehicle for most if not

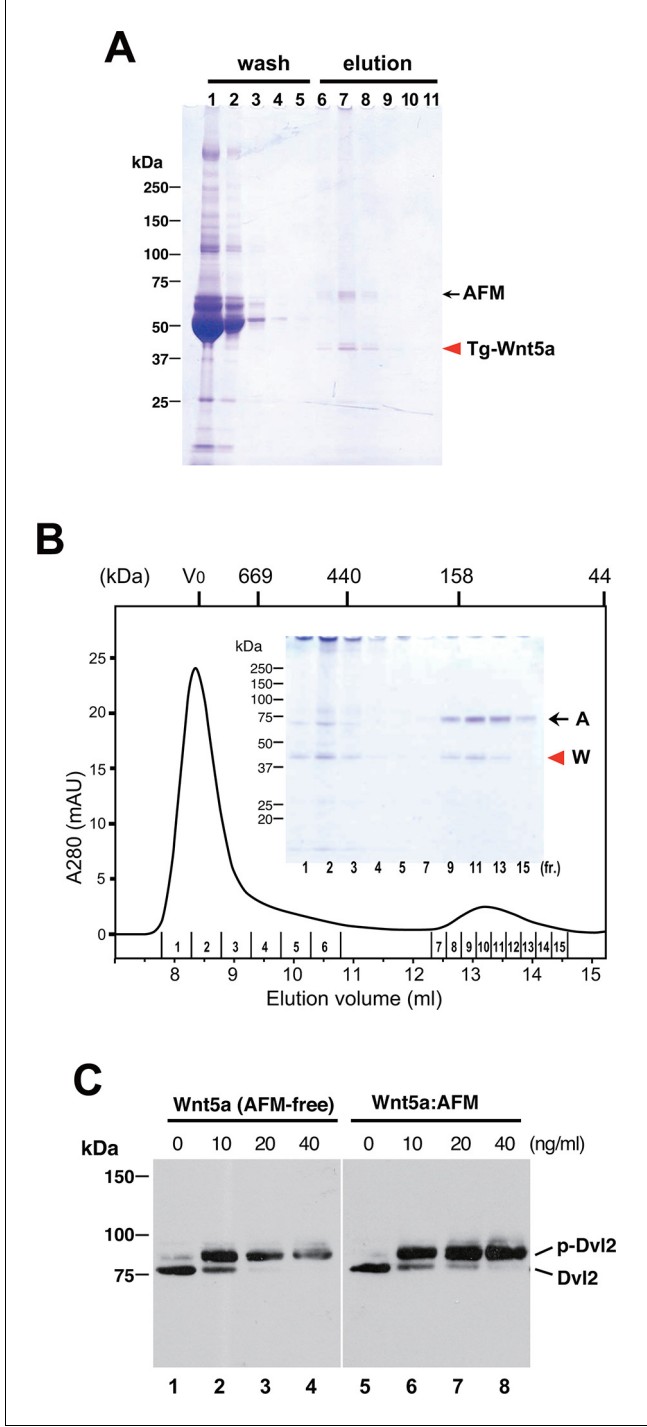

**Figure 6.** Serum AFM binds and solubilize Wnt5a. (**A**) Purification of Tg-Wnt5a from the CM of stable HEK cell line was conducted as in the case for Tg-Wn3a, and the resulting column fractions were analyzed by SDS-PAGE and Coomassie Blue staining. Positions for the 70-kDa bovine AFM and the 40-kDa mouse Wnt5a are shown at the right. (**B**) SEC profile of the tag-purified Wnt5a. Note that the ~150-kDa peak (fractions 9–13) contains roughly equimolar amounts of AFM (A) and Wnt5a (W) proteins as judged by the non-reducing SDS-PAGE (*inset*). The 40-kDa Wnt5a was present in both HMW and monodispersed fractions. See also *Figure 6—figure supplement 1*. (**C**) Biological activity of AFM-Wnt5a complex. NIH3T3 cells were incubated with either AFM-free, untagged Wnt5a (lanes 1–4) or the purified Tg-Wnt5a:AFM complex (lanes 5–8) at increasing concentrations. The level of cellular Dvl2 phosphrylation was assessed by the mobility shift of Dvl2 band in the anti-Dvl2 immunoblots using the whole cell lysates.

*Figure 6 continued on next page*

*Figure 6 continued*

The following figure supplement is available for figure 6:

**Figure supplement 1.** Identification of the 40-kDa band as Wnt5a.

all Wnt proteins to aid their simple purification. The culture media containing various Wnts in complex with AFM thus obtained were tested for their ability to activate Wnt/β-catenin signaling in TCF reporter cells. As shown in *Figure 7D*, only Wnt3 and Wnt3a showed significant activity in this assay system. This result contradicts with that obtained by Najdi et al who reported that, in addition to Wnt3 and Wnt3a, other Wnts including Wnt1, Wnt7a, Wnt7b, Wnt8a, Wnt9b, and Wnt10b were capable of activating Wnt/β-catenin signaling in HEK cells to some extent (*Najdi et al., 2012*). However, they evaluated the activity of untagged Wnts in an autocrine format by directly transfecting each Wnt into the reporter cells rather than exogenously adding the secreted Wnt proteins. We confirmed that some of our PA-tagged Wnts including Wnt1, Wnt9b, and Wnt10b also showed some activity when tested in the autocrine type of assay (*Figure 7—figure supplement 2*). It is therefore possible that the discrepancy reflects the requirement of the cell-associated Wnt presentation for some Wnt subtypes to signal, which has been postulated for Wnt1 (*Green et al., 2013*). Still, there is a possibility that the signaling property of these Wnts are attenuated by the complex formation with AFM.

Taking advantage of the scalable nature of the transient expression using suspension cells, we performed large-scale expression/purification of mouse Wnt3a and human Wnt3. As shown in *Figure 8A*, both Wnt proteins can be purified from the culture media of transfected Expi293F cells using anti-PA tag immunoaffinity resin, solely in complex with the co-expressed human AFM. Typically, we obtained ~170 µg mouse Wnt3a and ~120 µg human Wnt3 from a 300 ml culture media. Although it is difficult to tell whether AFM and Wnt can form complex before the secretion (e.g., in the secretary compartments) or the complex formation is exclusively a post-secretion event (i.e., associated after both proteins are secreted separately), this result indicates that AFM does not need to be added as protein but can be simultaneously produced with Wnts in the same cells to support its secretion. As in the case of AFM-Wnt complex formed in the serum-containing culture media (i.e., *Figures 2* and *6*), the AFM-Wnt3a/3 complex produced from cells upon co-expression primarily behaved as a monodispersed molecular species (*Figure 8B,C*, fraction II). Furthermore, both Wnts are not present in the HMW peak (fraction I), suggesting that the absence of serum components in the culture media is advantageous to obtain cleaner Wnt samples devoid of contaminating proteins such as HDL. Although the estimated molecular size for these complexes (~140 kDa) is significantly different from that obtained in the previous experiment (100 kDa, *Figure 4A*), the recombinant human AFM used in the co-expression experiments carries one more N-glycosylation site and a long (36 residues) N-terminal tag peptide compared to the serum-derived, natural bovine AFM, which may have contributed to the larger Stokes radius of the complex. Finally, we confirmed that AFM-Wnt3a as well as AFM-Wnt3 complexes thus produced were active in the Wnt/β-catenin signaling reporter assay, exhibiting similar concentration dependency with that of AFM-Wnt3a complex purified from serum-containing CM (*Figure 8D,E*). Overall, the AFM co-expression strategy combined with the use of transient transfection of suspension cells is an excellent choice of method when the production of biologically active Wnt ligands at large quantity is required.

## Discussion

Despite their fundamental involvement in general metazoan development as well as various human diseases, Wnt proteins and their mechanism of action had evaded biochemical scrutiny for many years. The most enigmatic character of Wnt proteins is their hydrophobic nature due to the covalent lipid modification discovered by Willert et al (*Willert et al., 2003*), which was later identified by Takada and colleagues as an acylation of conserved Ser residue with palmitoleic acid (*Takada et al., 2006*). Although the Wnt purification method established by Willert et al set an important milestone in the Wnt research, its application to the structure-function studies had been slow due to both the technical difficulties associated with the method that involves multiple steps of large-scale

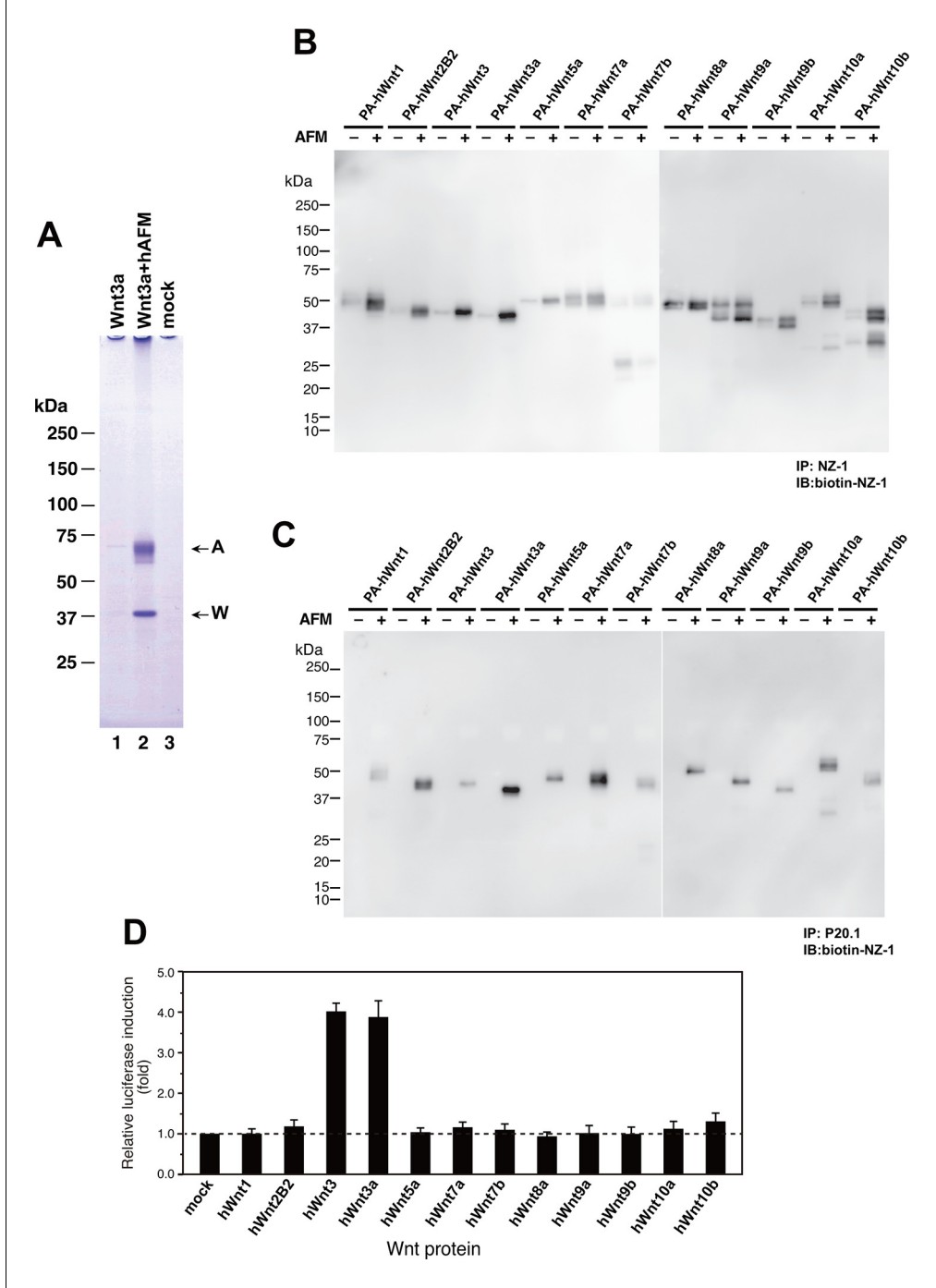

**Figure 7.** Increased production/secretion of various Wnt proteins by the co-expression with AFM. (**A**) Expi293F cells were transiently transfected with PA-tagged mouse Wnt3a (lane 1), PA-tagged mouse Wnt3a + Tg-tagged human AFM (lane 2), or mock (lane 3) plasmids under the serum-free condition and the resultant culture supernatants were subjected to the immunoprecipitation with anti-PA tag antibody NZ-1 and analyzed on a nonreducing 5–20% SDS-PAGE gel followed by Coomassie Blue staining. A, Tg-hAFM; W, PA-mWnt3a. (**B–D**) AFM co-expression facilitates various Wnt secretion. N-terminally PA-tagged human Wnts are transiently co-transfected with Tg-tagged human AFM into Expi293F cells, and the culture media are immunoprecipitated with either anti-PA NZ-1 (**B**) or anti-Tg P20.1 (**C**), followed by immunoblotting with biotinylated NZ-1 to visualize PA-tagged Wnts. In (**D**), the same set of culture media are subjected to the TCF reporter assay as in *Figure 1B*, to see if they can induce Wnt/β-catenin signaling. The data are mean ± SE (n = 4) from a representative experiment. See *Figure 7— source data 1*.

*Figure 7 continued on next page*

*Figure 7 continued*

The following source data and figure supplements are available for figure 7:

**Source data 1.** The Excel spreadsheet source file for *Figure 7D*.

**Source data 2.** The Excel spreadsheet source file for *Figure 7—figure supplement 2*.

**Figure supplement 1.** Expression/secretion profile of all human Wnt constructs in the presence of serum.

**Figure supplement 2.** Signaling activity of 19 PA-tagged Wnts in an autocrine type assay.

chromatography, and the unavoidable presence of detergent in the final sample. The next breakthrough in the Wnt preparation was brought by Garcia and colleagues, who determined the crystal structure of Xenopus Wnt8 (*Janda et al., 2012*). They expressed Wnt8 protein as a stable and tight complex with a fragment of its receptor Frizzled8 (Fz8), which sequestered the problematic acyl chain and increased the solubility of the complex. The structure clarified the essential role of the fatty acid moiety in the receptor recognition, and offered a first molecular view about how Wnt initiate signal outside the cell. However, this kind of soluble Wnt proteins cannot be used in the functional assays, because the important active site (i.e., the receptor binding site) is already masked in the complex and unavailable. Another way of making Wnt proteins soluble in aqueous solution is to remove the acyl chain that renders the insolubility. In fact, Kakugawa et al have recently reported an extracellular deacylase called Notum, which removes the palmitoleic acid attached to the conserved Ser residue in Wnt3a (*Kakugawa et al., 2015*). However, the resultant deacylated Wnt, although it may be water-soluble, lacks biological activity and its utility in the structure-function analysis is limited. Probably the best known method to present purified active Wnt proteins in the absence of potentially cell toxic compounds is to reconstitute them in a liposome (*Morrell et al., 2008*). Although the liposome-packaged Wnt3a have long life at 37°C and may potentially be suitable for the in vivo delivery (*Dhamdhere et al., 2014*), one still needs to produce large quantity of CHAPS-solubilized Wnt proteins, which remains technically demanding.

We discovered that AFM has unexpected ability to solubilize/partition the lipophilic Wnt proteins in the culture media of Wnt-producing cells. AFM can specifically and stoicheometrically interact with several Wnt proteins, allow them to stay dispersed in aqueous solution, and keep them from inactivation/aggregation during the storage. As AFM is an endogenous protein abundantly present in mammalian body, complexation with AFM is an ideal format of Wnt preparation suitable for the use in various biochemical, cell biological, or even therapeutic applications. AFM is a member of albumin superfamily plasma proteins, which comprises of four proteins existing in blood at widely different concentrations ranging from as low as 50 ng/ml for AFP to extraordinarily high 40 mg/ml for serum albumin (*Lichenstein et al., 1994*). AFM is present in normal adult blood at 10–30 µg/ml but its physiological function remains unknown, except for its potential role as a vitamin E transport protein (*Dieplinger and Dieplinger, 2015*). Interestingly, AFM is able to bind hydrophobic substances like other albumin family proteins, but the binding specificity seems to be unique (*Voegele et al., 2002*). The facts that 1) only AFM but not other three albumin family proteins can capture Wnts and 2) the interaction between AFM and Wnt is completely disrupted by 1% CHAPS seem to indicate that AFM uses its unique lipid-binding pocket to bind and sequester the palmitoleic moiety of Wnt proteins. If such lipid binding specificity of AFM is responsible for this special ability, all Wnt ligands, or even non-Wnt proteins that are modified by palmitoleic acid may be protected by AFM. In fact, we found that 12 different Wnt proteins were able to associate with AFM, leading to the secretion from cells into a serum-free medium upon co-expression with AFM. The idea that AFM binds and sequester fatty acid chain of Wnts may seem to contradict with the fact that AFM-Wnt complex is highly active in cell-based assays, because the acyl chain moiety has to be available for the receptor (i.e., Fz) binding to initiate signaling. We speculate that the affinity of AFM to the fatty acid is strong enough to make stable complex with lipid-modified Wnt but weaker than that of Fz CRD, enabling it to 'deliver' the ligand to the receptor at the cell surface.

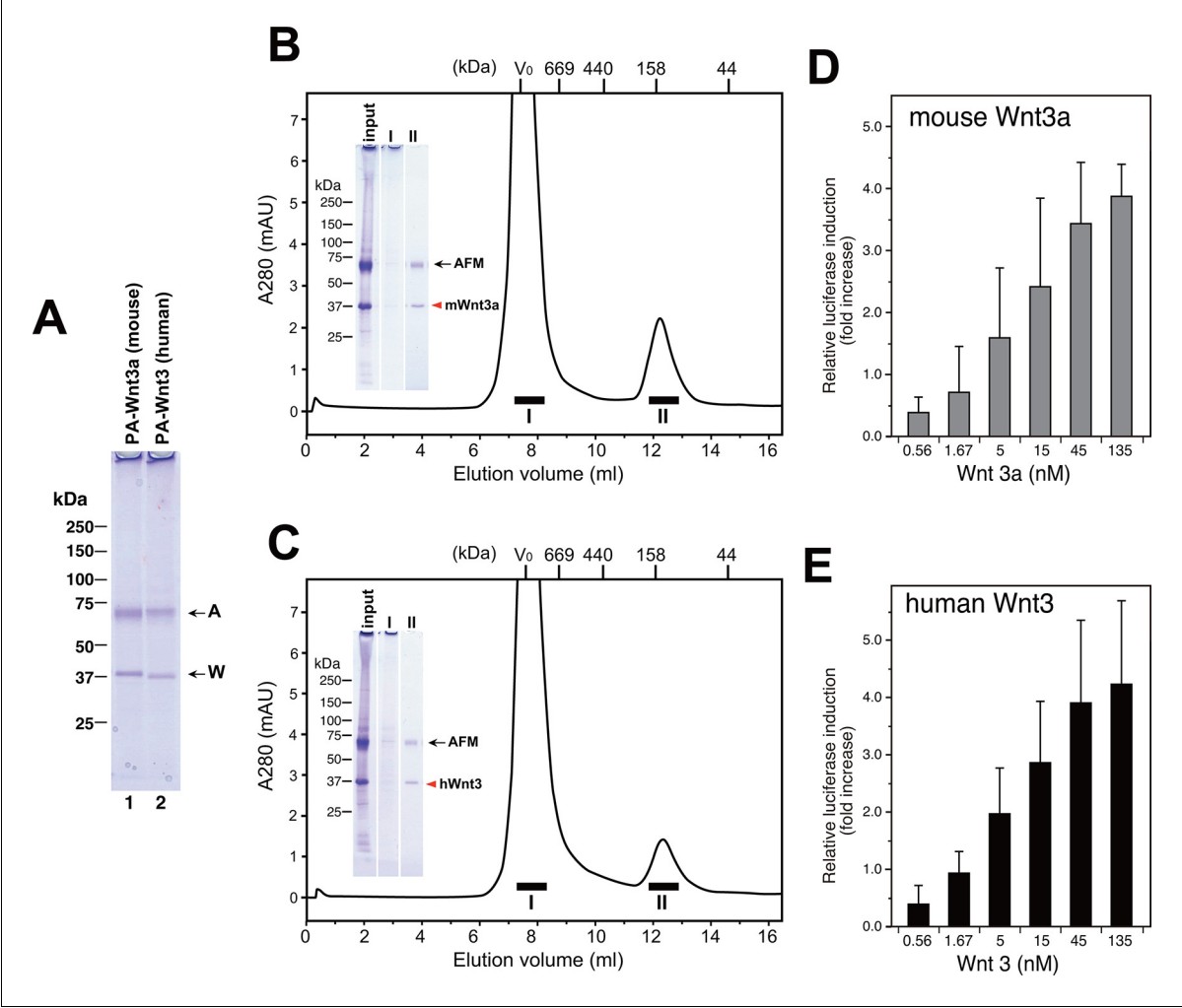

**Figure 8.** Large scale production of Wnt3a and Wnt3 by using AFM co-expression method. (**A**) A 300 ml culture of Expi293F cells were used to co-express Tg-tagged human AFM and PA-tagged mouse Wnt3a or human Wnt3. The AFM-Wnt complexes were purified from the co-transfected culture media using NZ-1-immunoaffinity resin and checked for its purity on a nonreducing 5–20% SDS-PAGE gel followed by Coomassie Blue staining. (**B,C**) SEC profile of the NZ-1-purified AFM-Wnt3a (**B**) or AFM-Wnt3 (**C**) complex. The eluted material from the NZ-1 immunoaffinity chromatography (input, same as the sample shown in (**A**)) was subjected to the SEC on Superdex 200, showing two peaks. The void peak (fraction I) comprised only of HMW aggregates that barely enter the top of separation gel, while the ~140-kDa peak (fraction II) contained AFM and Wnt3a/Wnt3 at similar amounts. (**D,E**) Signaling activity of AFM-Wnt complex prepared by AFM co-expression method. The NZ-1-purified AFM-Wnt3a (**D**) or AFM-Wnt3 (**E**) complex was subjected to the TCF reporter assay as in **Figure 5A**. The data are mean ± SD of three independent experiments, in which quadruplicate determinations were made. See **Figure 8—source data 1**.

The following source data is available for figure 8:

**Source data 1.** The Excel spreadsheet source file for **Figure 8D and E**.

Although the experimental/technological importance of the activity of AFM as a Wnt carrier protein is clear, its relevance in physiology remains to be explored. AFM, as other albumin family proteins, is primarily synthesized in the liver and present not only in blood but also in other adult body fluids including cerebrospinal, ovarian follicular, and seminal fluids (***Dieplinger and Dieplinger, 2015***). However, nothing is known about its expression pattern or potential function during mammalian development. As invertebrates do not have albumin family proteins, AFM does not seem to play essential roles in the evolutionary ancient function of Wnt singaling pathway. Nevertheless, the Wnt-binding activity is likely to be conserved among mammalian AFM, since bovine (***Figure 3A***), human (***Figures 3B and C***), and mouse (data not shown) AFM are all capable of supporting Wnt secretion

from cells. Brack et al reported that Wnt or Wnt-like molecule present in the serum from aged mice may be responsible for the conversion of muscle stem cells from myogenic to fibrogenic phenotypes (*Brack et al., 2007*). It is possible that Wnt ligands present in adult body fluids play unidentified (patho)physiological roles under certain condition, which may be regulated by AFM through its carrier function.

In *Drosophila*, there is a known Wnt carrier protein called Swim, which is a secreted glycoprotein of ~50 kDa belonging to the lipocalin family transport proteins (*Mulligan et al., 2012*). Like AFM, Swim binds Wg ligand in an acyl chain-dependent manner, and maintains the solubility and activity of purified Wg. Reduction of Swim leads to a more confined distribution of extracellular Wg around the producing cells, suggesting its role in establishing the proper long-range Wg gradient. There are a number of other factors that can mediate transport of Wnt ligands from the secreting cells to the responding cells, including Wnt-containing exosome assembled by the Evi/Wntless-mediated Wnt trafficking (*Gross et al., 2012*), large lipoprotein particles that carry not only Wnt but also other lipid modified morphogen Hedgehog (*Panakova et al., 2005*), and cell surface heparan sulfate proteoglycans that serve as the reservoir for Wnt to shape the extracellular gradient (*Han et al., 2005*). AFM may be considered as a new candidate factor involved in the regulation of Wnt distribution and presentation in mammals. Genetic manipulation of AFM gene in mice and their phenotypic analyses are eagerly waited.

## Materials and methods

### Plasmids and cell culture

cDNAs coding for mouse Wnt3a and Wnt5a, monoclonal antibodies against mouse Wnt3a and Wnt5a, and L cell stably expressing mouse Wnt3a (L-3a) (*Shibamoto et al., 1998*) were all kindly provided by Shinji Takada (Okazaki Integrated Bioscience Center, Japan). The Open Source Wnt Project plasmids containing ORF clones for human Wnt proteins (*Najdi et al., 2012*) were obtained from Addgene. A cDNA coding for the human afamin was a generous gift from Luc Bélanger (Research Center, L'Hotel-Dieu de Québec, Canada). cDNAs coding for the following proteins were obtained by DNA synthesis based on their reported nucleotide sequences; bovine AFM (NM_001192175), human serum albumin (BC034023), human AFP (BC027881), and mouse VDBP (AK010965). A highly pure, carrier-free recombinant mouse Wnt3a protein was purchased from R&D Systems (catalogue No. 1324-WNP-010/CF). HEK293T, HEK293S GnT1 (kindly provided by G. Korhana), and L-3a cells are maintained in basal media containing DMEM (for HEK lines) or DMEM/F12_1:1 (for L-3a), supplemented with 10% fetal calf serum (FCS). All cell transfection experiments were performed using X-tremeGENE HP (Roche).

### Recombinant production of Wnt proteins

Expression constructs for mouse and human Wnt proteins with various N-terminal tags were designed to include tobacco etch virus (TEV) protease cleavage sequence after the tag as shown in *Supplementary file 1*, and constructed by extension PCR. Expression plasmids for various albumin family proteins with N-terminal tag (either Tg or PA tag) were also constructed by the same strategy. The coding region for Tg-Wnt3a was subcloned into a mammalian episomal expression vector pEB-Multi-Hyg (Wako Pure Chemical, Japan) and used to transfect HEK293S GnT1⁻ cells. Transfected cells were selected against medium containing 0.2 mg/ml hygromycin B, and the resistant polyclonal cell population confirmed to maintain high Wnt3a-producing ability over the repeated passages were used for the protein production. HEK293S GnT1⁻ cells stably producing Tg-Wnt5a were established in a similar manner. For the co-expression of Wnt and AFM, expression plasmids for Tg-hAFM and N-terminally PA tagged mouse or human Wnts were co-transfected at a DNA ratio of 10:1 using the Expi293 Expression System (Life Technologies Inc. Tokyo Japan), according to the procedures recommended by the manufacturer. The culture medium was harvested at 90 hr post-transfection, and immediately used for the Wnt/β-catenin reporter assay or the protein purification.

### Protein purification

Purification of the TARGET-tagged Wnt proteins was conducted according to the general procedure described previously (*Nogi et al., 2008*). Briefly, precleared CM was incubated with P20.1-

Sepharose at 1~2 ml beads/100 ml CM at 4°C for 3 hr, followed by packaging into a small column. The resin was washed with 5 column volume of 20 mM Tris, 150 mM NaCl, pH 7.5 (TBS) and then eluted with 0.2 mg/ml C8 peptide (PRGYPGQV) in TBS. Elution peak fractions were combined and concentrated by a centrifugal ultrafiltration devise (Spin-X UF, MWCO 10,000, Corning). For further purification, if necessary, the sample was subjected to size exclution chromatography on Superdex 200 10/300GL column (GE Healthcare) equilibrated with TBS. When used in cell-based activity assays, purified samples were dialyzed against phosphate-buffered saline (PBS) and filter-sterilized before use. PA-tagged albumin family proteins were produced using the Expi293 Expression System, and purification of these proteins as well as the Tg-AFM:PA-Wnt complexes from the CM using NZ-1-Sepharose affinity chromatography was performed based on the protocol reported by Fujii et al (*Fujii et al., 2014*). The concentration of the Wnt3a or Wnt3 in the purified AFM complex was estimated by a densitometric analysis of protein bands in the SDS-PAGE gel using BSA as standard. Untagged Wnt5a was purified from the CM of stable L cell line as described previously (*Kurayoshi et al., 2007*).

## Antibodies

Monoclonal antibodies against bovine AFM were prepared by a conventional hybridoma method using the mouse myeloma cell line P3U1 and Balb/C mice immunized with the tag-cleaved bovine AFM protein. Of total of six independent clones, a clone B91 (IgG1, κ) recognized native bovine AFM with high affinity and was compatible with the immunoprecipitation experiments (see below). The same bovine AFM protein was used as immunogen to raise antisera in rabbits. To raise antisera against bovine ApoA1, two peptide segments (residues 1–12, DDPQSSWDRVKD, and residues 161–174, QLAPYSDDLRQRLT) were selected based on their potential antigenicity and used as immunogens after conjugation with KLH. Antibodies were purified from culture supernatants (for monoclonal antibodies) or rabbit sera (for polyclonal antibodies) by using affinity chromatography on Protein A-Sepharose (GE Healthcare). Purified anti-TARGET tag P20.1 and anti-PA tag NZ-1 antibodies (*Fujii et al., 2014*; *Nogi et al., 2008*) were obtained from Wako Pure Chemical Co.

## Immunoprecipitation and immunoblotting

Purified B91 IgG was coupled to CNBr-activated Sepharose beads (GE Healthcare) at a density of ~2 mg IgG/ml beads by the protocol provided by the manufacturer. For the immunoprecipitation of bovine AFM from the serum-containing media, a 20 µl of B91-Sepharose beads were incubated with 1 ml of various CM at 4°C for 3 h, washed three times with TBS, and boiled after the addition of 30 µl of SDS sample buffer. The samples were subjected to SDS-PAGE and visualized by either Oriole fluorescent protein stain (BIO-RAD) or anti-Wnt3a immunoblotting. For anti-Wnt3a and anti-Wnt5a immunoblottings, the samples were separated by SDS-PAGE, transferred to PVDF membrane, and incubated with 5 µg/ml anti-Wnt antibodies that had been biotinylated by using EZ-Link NHS-LC Biotin (PIERCE Chemical Co.). After washing, the membranes were probed with peroxidase-conjugated Streptavidin (VECTOR Laboratories, SA-5004), followed by visualization with enhanced chemiluminescence reagent. All other immunoblotting experiments were conducted in a similar manner, except for the use of unlabeled primary antibodies (30 µg/ml for rabbit polyclonal anti-bovine AFM, 1/500 dilution for rabbit antisera against ApoA1) and peroxidase-conjugated secondary anti-rabbit IgG (Sigma A-6154, 0.4 µg/ml). Pull-down experiments of various Wnt proteins co-expressed with Tg-hAFM in Expi293F cells were conducted using NZ-1- or P20.1-Sepharose as described previously (*Fujii et al., 2014*; *Nogi et al., 2008*), followed by biotinylated NZ-1 immunoblotting to visualize PA-tagged Wnts.

## Wnt activity measurements

For the luciferase reporter assays, HEK293 cells stably integrated with a firefly luciferase gene under the control of TCF/LEF response element (BPS Bioscience) were used. Cells were suspended at ~8 $10^5$ cells/ml and seeded in 96-well plates at 50 µl/well, followed by an overnight culture before the assay. For the preparation of CM containing Wnt3a, stable Wnt3a-producing cells were cultured for 3~4 days until they reach 80–90% confluency. The harvested CM were diluted with the control CM from the similarly cultured HEK293T cells to adjust for the presence of any common cell metabolites. When purified Wnt3a or AFM-Wnt3a complex were used, they were diluted in PBS from the stock

solution and further diluted with a fresh DMEM containing 10% FCS. About 50-µl of the test reagents were added to the reporter cells and incubated for 6 prior to a brief washing and cell lysis. Reporter activities contained in the lysates were measured using a dual luciferase reporter assay system (Promega). Each assay was performed in quadruplicate.

For the Dvl2 phosphorylation assay, NIH3T3 cells were cultured in the presence of varying concentrations of Wnt5a proteins. As the stock solution of the purified untagged Wnt5a contained 1% CHAPS (*Kurayoshi et al., 2007*), AFM-Wnt5a complex devoid of detergents were preincubated with 1% CHAPS for 1 hr at 4℃. Both samples were diluted with the fresh media containing 1% BSA, where the final concentration of CHAPS was below 0.01%. After the treatment with Wnt5a for 2 hr, the cells were lysed in lysis buffer (20 mM Tris-HCl [pH 8.0], 1% Nonidet P-40, 137 mM NaCl, 10% glycerol, 1 mM phenylmethylsulfonyl fluoride, 20 µg/ml aprotinin, 20 µg/ml leupeptin, 5 mM NaF, 5 mM $Na_3VO_4$, and 50 mM β-glycerophosphate) and cell lysates were subjected to the immunoblotting with anti-Dvl2 antibody (Cell signaling Technology).

## Human intestinal organoid culture assay

The supporting effect of Wnt3a on the growth and expansion of human intestinal organoid cultures were evaluated using the method established previously (*Sato et al., 2011*). Colonic samples were obtained at Keio University Hospital with written informed consent and consent to publish. Normal mucosa samples were derived from tissue biopsy specimen taken from patients with colonic tumor, which were located at least 5 cm away from the edge of tumors. Crypts were dissociated from the samples by a combination of EDTA treatment and physical pipetting, and isolated. They were suspended in Matrigel, placed at the center of each well of a 48-well plate, and overlaid with 125 µl of defined serum-free medium containing a cocktail of essential factors including R-spondin (500 ng/ml), EGF (50 ng/ml), Noggin (100 ng/ml), A83-01 (TGF-β receptor 1 inhibitor, 500 nM), and SB 202190 (p38 MAPK inhibitor, 10 µM). The culture was started by the addition of the same volume of Wnt3a-containing solution, either as a fresh CM of L-3a cells or concentration-adjusted Wnt3a or AFM-Wnt3a complex in PBS, and the crypt morphology and numbers were observed daily. Upon the outgrowth, the organoids were passaged every week by gentle pipetting at 1:5 ratio into the fresh medium.

## Acknowledgements

We would like to thank Ryoko Asaki for excellent technical assistance and Mie Sakai for preparation of the manuscript. This work was supported in part by the Grant-in-Aid for Scientific Research on Innovative Areas (Analysis and Synthesis of Multidimensional Immune Organ Network, to JT) from the Ministry of Education, Culture, Sports, Science and Technology of Japan (MEXT), by the 'Platform for Drug Discovery, Informatics, and Structural Life Science'grant from the MEXT (to JT), by the X-ray Free Electron Laser Priority Strategy Program"' grant from the MEXT (to JT), and by the Research Center Network for Realization of Regenerative Medicine project from Japan Agency for Medical Research and Development (AMED) (to TS).

## Additional information

### Funding

| Funder | Grant reference number | Author |
| --- | --- | --- |
| Japan Agency for Medical Research and Development | Research Center Network for Realization of Regerative Medicine | Toshiro Sato |
| Ministry of Education, Culture, Sports, Science, and Technology | Grant-in-Aid for Scientific Research on Innovative Areas | Junichi Takagi |
| Ministry of Education, Culture, Sports, Science, and Technology | Platform for Drug Discovery, Informatics, and Structural Life Science Grant | Junichi Takagi |

| Ministry of Education, Culture, Sports, Science, and Technology | X-ray Free Electron Laser Priority Strategy Program | Junichi Takagi |

The funders had no role in study design, data collection and interpretation, or the decision to submit the work for publication.

## Author contributions
EM, AK, TS, Acquisition of data, Analysis and interpretation of data, Drafting or revising the article; HH, HY, KTK, MM, Acquisition of data, Analysis and interpretation of data; JT, Conception and design, Acquisition of data, Analysis and interpretation of data, Drafting or revising the article

## Author ORCIDs
Junichi Takagi, http://orcid.org/0000-0002-1219-475X

## Ethics
Human subjects: Colonic samples were obtained at Keio University Hospital with written informed consent and consent to publish. Normal mucosa samples were derived from tissue biopsy specimen taken from patients with colonic tumor, which were located at least 5 cm away from the edge of tumors. The study was approved by the ethical committees of the Keio University Hospital.

## Additional files

### Supplementary files
• Supplementary file 1. Construct design of tagged Wnt proteins. Amino acid sequence of the N-terminal regions for the all tagged Wnt protein constructs used in this study starting at the initiation Met are shown.

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
