## [Decision Letter]

Thank you for submitting your work entitled "Active and water-soluble form of lipidated Wnt protein is maintained by a serum glycoprotein afamin/α-albumin" for consideration by *eLife*. Your article has been favorably reviewed by John Kuriyan (Senior Editor) and three reviewers, one of whom is a member of our Board of Reviewing Editors.

The reviewers have discussed the reviews with one another and the Reviewing editor has drafted this decision to help you prepare a revised submission.

Summary:

Purification of Wnt proteins has been difficult due to their generally poor expression and the presence of a covalently attached lipid. The presence of detergent needed for solubility makes these preparations incompatible with some applications. Moreover, the purified proteins are not very stable, and their activity is also somewhat variable, making assessment of specific activity impossible. Here, the authors show that afamin/α-albumin forms 1:1 complexes with two different Wnt proteins. These complexes are highly stable, do not require any detergent, and are bioactive.

Essential revisions:

The reviewers agree that this work potentially represents a significant technical advance in purifying active Wnt proteins. However, only two Wnt proteins have been tested, and these two are known to be relatively easy to express and purify. Therefore, it is felt that you need to demonstrate that afamin can aid in the expression and purification of at least a few other Wnt proteins. Although there may not be suitable assays for Wnts other than 3a and 5a, Wnt10 is likely to activate β-catenin and would therefore be a useful candidate for assessing functionality.

The other important issue to address is the claim of higher specific activity for the Wnt3a/afamin complex versus the standard purification, which would be an important advantage. However, the statement that the EC50 value is "nearly 10 fold lower" than the purified commercial Wnt 3a does not appear to be supported by the data in Figure 5, as it is not clear where saturation is reached for either graph, especially that of the R&D material.

[Editors' note: further revisions were requested prior to acceptance, as described below.]

Thank you for resubmitting your work entitled "Active and water-soluble form of lipidated Wnt protein is maintained by a serum glycoprotein afamin/α-albumin" for further consideration at *eLife*. Your revised article has been favorably evaluated by John Kuriyan (Senior Editor) and a Reviewing editor. The manuscript has been improved but there are some remaining issues that need to be addressed before acceptance, as outlined below:

The revised manuscript has largely addressed the reviewer concerns, and now shows that afamin can aid in the expression of a number of Wnt proteins. You have also demonstrated improved purification and demonstrated activity of another Wnt, Wnt3. It is not clear that the expression levels offer a clear advantage versus serum culture for most Wnts (in this regard, have you reprobed the blot in Figure 7—figure supplement 1 for the tag?), but the simplicity of the purification seems to offer a real advantage in time and cost.

It is troublesome that no other afamin-stabilized Wnts give a TOPFLASH signal in Figure 7, given the Najdi et al. results. Although you speculate that this is due to using a paracrine rather than autocrine assay, have you assessed this by co-transfection? You should address this question before publication.

---

## [Author Response]

*The reviewers agree that this work potentially represents a significant technical advance in purifying active Wnt proteins. However, only two Wnt proteins have been tested, and these two are known to be relatively easy to express and purify. Therefore, it is felt that you need to demonstrate that afamin can aid in the expression and purification of at least a few other Wnt proteins. Although there may not be suitable assays for Wnts other than 3a and 5a, Wnt10 is likely to activate β-catenin and would therefore be a useful candidate for assessing functionality.*

We thank the reviewers for recognizing the merit of our findings. According to the request above, we obtained DNAs for all human Wnt proteins made public by The Open Source Wnt Project and constructed PA-tagged expression plasmids for all of them. For the 12 Wnts (out of 19) that could be secreted from transfected HEK cells in the presence of serum (now shown in Figure 7—figure supplement 1), we tested afamin co-expression and found that all of them show moderate to substantial increase in the expression/secretion into the media (new Figure 7). Moreover, all of them were found to be associated directly with afamin (new Figure 7), indicating that afamin can indeed aid in the expression and secretion of a wide range of Wnt subtypes other than Wnt3a and Wnt5a. When we performed TCF reporter assay for these samples, however, only Wnt3 and Wnt3a were active (new Figure 7). Some of these Wnts are reported to be capable of activating β-catenin pathway, as pointed out in the review comment, and we do not know the exact reason for this discrepancy. One thing we do know is that most studies (including Najdi et al. paper reporting the construction of the Open Source Wnt collection) use “autocrine” type of experiment where each Wnt is directly transfected into the reporter cell itself. So we speculate that certain Wnts can activate β-catenin pathway efficiently only when they are simultaneously expressed in the same cell. As Wnt3 was the only additional Wnt whose activity can be evaluated in our system, we chose this Wnt to see if the list of Wnt types with confirmed afamin-aided purification can be expanded. This was indeed the case, because we could purify highly active human Wnt3 in complex with human afamin from the co-transfected HEK cells, just as easy as for mouse Wnt3a (new Figure 8). These additional data are assembled into new Figure 7 and Figure 8, and the results are described in the Results section in the revised manuscript. Due to the increased number of the Wnt expression constructs (now 22 in total), the schematics of the construct design for all Wnt proteins are made and submitted as a [Supplementary-material SD8-data] (replacing the old Figure 1–figure supplement 1).

The other important issue to address is the claim of higher specific activity for the Wnt3a/afamin complex versus the standard purification, which would be an important advantage. However, the statement that the EC50 value is "nearly 10 fold lower" than the purified commercial Wnt 3a does not appear to be supported by the data in Figure 5, as it is not clear where saturation is reached for either graph, especially that of the R&D material.

The reason why we used these rather vague words to describe the activity difference between the two samples was, as correctly pointed out by the reviewer, the fact that we were not sure if the activity of the R&D sample has reached saturation. Also, the dose-dependent curve for the AFM-Wnt3a complex did not conform to a standard sigmoidal shape, making it difficult to clearly define the EC50 values. Another technical excuse is that our TCF reporter assay was quantitatively not so reproducible despite our best efforts, making us hesitant to give exact values. We repeated the experiment using more reliable assay system (described in more detail in the following section) and obtained highly reproducible dose-dependent curves for both samples (now appears as a new Figure 5). It is confirmed that the activity for the AFM-Wnt3a complex has reached saturation, with the EC50 being somewhere between 1.67 and 5 nM. The activity of R&D sample on the other hand still cannot be saturated within the concentration range used. We did not want to increase the concentration further not only because it becomes unbearably expensive (requiring 7 µg of ~$2,000/10µg protein for additional one data point) but also it causes inclusion of higher concentration of CHAPS brought into the cell culture. If we can assume that the maximum signal elicited by the two different sample is identical (which is theoretically reasonable), it is safe to assume that the EC50 for the R&D sample is between 15 and 45 nM. Based on these considerations, we decided to restate the description of the difference as “5~10-fold lower”, which is a conservative estimate. These points are all described clearly in the revised manuscript (Results, fourth paragraph).

[Editors' note: further revisions were requested prior to acceptance, as described below.]

*The manuscript has been improved but there are some remaining issues that need to be addressed before acceptance, as outlined below: The revised manuscript has largely addressed the reviewer concerns, and now shows that afamin can aid in the expression of a number of Wnt proteins. You have also demonstrated improved purification and demonstrated activity of another Wnt, Wnt3. It is not clear that the expression levels offer a clear advantage versus serum culture for most Wnts (in this regard, have you reprobed the blot in Figure 7—figure supplement 1 for the tag?), but the simplicity of the purification seems to offer a real advantage in time and cost.*

We are glad that the reviewers felt that our revised manuscript sufficiently addressed most of the points raised and was improved. Regarding Figure 7—figure supplement 1, the purpose of having this figure was just to show which Wnt types were compatible with the serum-aided secretion after the N-terminal PA tagging (it *is* a Western blot probed with anti-PA tag antibody). We are not trying to say (and do not believe) that the expression levels upon the co-expression of afamin in the serum culture condition is higher than in the absence, because there is enough afamin protein already present in the bovine serum. In any case, we completely agree that the primary advantage of the current method over serum culture is the ability of obtaining the detergent-free Wnt proteins with a very simple protocol. Therefore, we revised the relevant sections in the text (Results, seventh and eighth paragraphs; and Discussion, second paragraph) to avoid misunderstanding that the afamin co-expression is superior to the serum-containing culture system as far as the expression level is concerned.

It is troublesome that no other afamin-stabilized Wnts give a TOPFLASH signal in Figure 7, given the Najdi et al.results. Although you speculate that this is due to using a paracrine rather than autocrine assay, have you assessed this by co-transfection? You should address this question before publication.

Yes, this point has been our concern too. Encouraged by the suggestion, we now performed autocrine assay (i.e., direct transfection of Wnts into the reporter cells) and found that similar set of Wnts (Wnt1, Wnt3, Wnt3a, Wnt9b, and Wnt10b) were capable of activating TCF reporter (new Figure 7—figure supplement 2). Although we do not know the exact reason why only Wnt3 and Wnt3a can signal in the paracrine-type assay format, it is possible that other Wnts (particularly Wnt1) need to be presented on cell surface for the efficient signaling. However, there is a possibility that these Wnts cannot signal when they are secreted into the medium as a complex with afamin. All these possibilities are stated in the revised manuscript (Results, seventh paragraph).